# Association between Exposure to Selected Heavy Metals and Blood Eosinophil Counts in Asthmatic Adults: Results from NHANES 2011–2018

**DOI:** 10.3390/jcm12041543

**Published:** 2023-02-15

**Authors:** Jun Wen, Mohan Giri, Li Xu, Shuliang Guo

**Affiliations:** Department of Respiratory and Critical Care Medicine, The First Affiliated Hospital of Chongqing Medical University, Chongqing Medical University, Chongqing 400016, China

**Keywords:** heavy metal, lead (Pb), eosinophil, asthma, National Health and Nutrition Examination Survey (NHANES)

## Abstract

(1) Background: Heavy metals are widely used and dispersed in the environment and people’s daily routines. Many studies have reported an association between heavy metal exposure and asthma. Blood eosinophils play a crucial role in the occurrence, progression, and treatment of asthma. However, there have thus far been few studies that aimed to explore the effects of heavy metal exposure on blood eosinophil counts in adults with asthma. Our study aims to discuss the association between metal exposure and blood eosinophil counts among asthmatic adults. (2) Methods: A total of 2026 asthmatic individuals were involved in our research from NHANES with metal exposure, blood eosinophils, and other covariates among the American population. A regression model, the XGBoost algorithm, and a generalized linear model (GAM) were used to explore the potential correlation. Furthermore, we conducted a stratified analysis to determine high-risk populations. (3) Results: The multivariate regression analysis indicated that concentrations of blood Pb (log per 1 mg/L; coefficient β, 25.39; *p* = 0.010) were positively associated with blood eosinophil counts. However, the associations between blood cadmium, mercury, selenium, manganese, and blood eosinophil counts were not statistically significant. We used stratified analysis to determine the high-risk group regarding Pb exposure. Pb was identified as the most vital variable influencing blood eosinophils through the XGBoost algorithm. We also used GAM to observe the linear relationship between the blood Pb concentrations and blood eosinophil counts. (4) Conclusions: The study demonstrated that blood Pb was positively correlated with blood eosinophil counts among asthmatic adults. We suggested that long-time Pb exposure as a risk factor might be correlated with the immune system disorder of asthmatic adults and affect the development, exacerbation, and treatment of asthma.

## 1. Introduction

Asthma is a heterogeneous disease characterized by chronic airway inflammation and respiratory symptoms [1,2]. It is one of the most common chronic respiratory disorders affecting approximately 358 million people worldwide [3,4], and 250,000 deaths are directly caused by asthma every year. The prevalence of asthma has increased in many countries in recent decades, and the prevalence of asthma in children and adults is estimated to be 8.1% and 7.9% in America, respectively [3,5]. According to a report, the US treatment and mortality costs for asthma have been estimated at USD 81 billion during 2008–2013 [6].

For many years, the presence of eosinophils in asthmatic inflammation has been recognized as an essential factor in the pathophysiology of the disease [7], and eosinophils are key inflammatory cells of “T2 high” asthma [8]. When monitoring patients with asthma, measurement of blood eosinophil counts is often used in clinical practice. As is known to us, the blood eosinophil count is a readily available biomarker, and existing research demonstrates that the blood eosinophil correlates reasonably well with sputum eosinophilia [9,10,11]. Among patients with asthma, elevated blood eosinophil counts have been associated with increased mortality [12] and are used to adjust treatment with corticosteroids, anti-IL-5, anti-IL-5-receptor-alpha or anti-IL-4 receptor alpha antibodies in asthma, resulting in improved asthma control and reduced exacerbation frequency [13,14,15,16,17,18,19,20]. Additionally, a high blood eosinophil count has been shown to be a risk factor for future asthma exacerbations in adults with persistent asthma [14,21,22,23]. In conclusion, blood eosinophils play a crucial role in the occurrence, progression, and treatment of asthma.

Heavy metals are widely used and dispersed in the environment, including the air, water, soil, dust, diet, and the manufacturing industry [24]. The general population is usually exposed to low concentrations of metals by ingesting contaminated water and food or inhaling environmental air pollution [25]. According to the Agency for Toxic Substances and Disease Registry, lead (Pb), cadmium (Cd), mercury (Hg), and so on are the top-priority contaminants. Heavy metals exposed to the environment may adhere to fine particles in the air and cause asthma as environmental allergens [26]. Previous studies observed that asthma induced by heavy metals was triggered by the immune system, indicating that heavy metals have great inflammatory potential and immunomodulatory effects on individuals [27,28]. Many studies have reported the association between heavy metals (Pb, Cd, Hg and Mn) and asthma [24,29,30,31,32,33].

In addition, some studies still demonstrated the correlation between common heavy metals (Pb, Hg, Cd, etc.) and blood eosinophil counts [34,35,36,37,38,39,40,41]. For example, previous studies have demonstrated that Pb exposure was implicated in alterations in humoral and cell-mediated immunity and the development of allergic conditions, including the production of serum IgE, eosinophil migration and proliferation, activation of Th2 cytokines, as well as an abnormality of bronchial responsiveness [37,38]. An experimental animal study in Iran indicated levels of total protein, total white blood cell counts, histamine, and eosinophils were elevated after guinea pigs inhaled aerosol Pb, which supported the proinflammatory effect of Pb poisoning [39]. Another experimental animal study demonstrated an elevation of circulating eosinophils after Cd exposure [40]. Additionally, a Korean study found Hg was not associated with blood eosinophil counts in 311 Korean children but was associated with increased blood lymphocyte counts [41]. Chao-Hsin Huang et al. conducted a study in which urine Mn was not associated with blood eosinophil counts among 2447 adults [34]. A small study involving 25 adult asthmatic patients and healthy subjects found that the asthma group had lower serum Se concentrations and higher indicators of oxidative stress [42].

However, there have been few studies that aimed to explore the effects of different kinds of exposure to heavy metals on blood eosinophil counts in adults with asthma so far. To explore the association between heavy metals (Pb, Cd, Hg, Se, and Mn) and blood eosinophil counts, in our study, we used a nationally representative sample of adults who participated in the 2011–2018 NHANES in the USA. We also comprehensively explored whether the association was different in various populations.

## 2. Materials and Methods

### 2.1. Data Source

The NHANES, which was conducted by the Centers for Disease Control and Prevention of America, collected information regarding the health and nutritional status of the US population every 2 years. NHANES used a complex, stratified sampling design, which can select representative samples of non-institutionalized civilians. The survey ethics review board of the CDC approved the NHANES procedures and protocols, and all participants provided written informed consent.

### 2.2. Study Population

Four cycles of NHANES data (2011–2012, 2013–2014, 2015–2016 and 2017–2018) were integrated into our study. These data included demographic data, examination data, laboratory data, and questionnaire data for the second analysis. A total of 39,156 participants were included in the NHANES from 2011 to 2018. We excluded individuals: (1) aged < 18 years old (n = 15331); (2) missing blood eosinophil data (n = 2147); (3) missing blood Pb, Cd, Hg, Se, and Mn data (n = 5573); (4) participants without asthma (n = 13673); (5) missing data about covariates in at least one of following (n = 406): education level, marital status, the ratio of family income to poverty, BMI, smoking status, hypertension history, diabetes history, blood urea nitrogen, blood creatinine and blood cotinine. Eventually, a large nationally representative sample (n = 2026) of American adults with asthma was enrolled in our study. The flow chart of the screening process is shown in Figure 1.

### 2.3. Measurement of Metal Exposures

The measurements of all the exposures of whole-blood lead, cadmium, mercury, selenium, and manganese were tested by inductively coupled plasma-dynamic reaction mass spectrometry (ICP-DRC-MS) on an ELAN 6100 DRC Plus or ELAN DRC II (PerkinElmer Instruments, Headquarters Office, 710 Bridgeport Ave., Shelton, CT 06,484–4794) at the CDC’s National Center for Environmental Health. Values of concentrations below the limit of detection (LOD) were imputed values of LOD/sqrt [2]. Detailed information on laboratory quality assurance and monitoring is available on the NHANES website.

### 2.4. Blood Eosinophil Count Measurement

Blood differential counts were performed in NHANES 2011–2018 using the Beckman Coulter HMX (Beckman Coulter, Fullerton, Calif), a quantitative and automated hematologic analyzer and leukocyte differential cell counter for in vitro diagnostic use in clinical laboratories. A detailed description of the laboratory methods can be found on the NHANES website.

### 2.5. Covariates and Asthma Assessment

Covariates were chosen a priori based on previous studies. Demographic data included gender, age (years old), race/ethnicity (Mexican American, other Hispanic, non-Hispanic white, non-Hispanic black, others), educational level (less than high school, high school, more than high school), poverty-to-income ratio, and marital status (married, single, living with a partner). Secondly, we also included examination data and personal life history data involving body mass index (kg/m^2^), smoking (smoked at least 100 cigarettes in life), diagnosis with hypertension (yes or no), and diagnosis with diabetes (yes, no, or borderline). Finally, variables of laboratory data included blood urea nitrogen (mmol/L), blood creatinine (mol/L), and blood cotinine (ng/mL). The assessment of asthma was based on the information from the questionnaire section of the US National Health Interview Survey. In order to assess asthma, participants were asked, “Has a doctor or other health professional ever told you that you have asthma?” If the participant responded “yes”, he or she was regarded as an asthma patient. A more detailed description of variables can be obtained on the NHANES official website (http://www.cdc.gov/nchs/nhanes/, 6 January 2023).

### 2.6. Statistical Analysis

According to the criteria of the CDC guidelines, we conducted a statistical analysis of blood metal concentration and blood eosinophil count. Blood metal concentration, blood eosinophil count, and other continuous variables were expressed as the mean and 95% CIs. The categorical variables were expressed in frequency or percentage. Firstly, we divided blood eosinophil count as a continuous variable into four quartiles. The weighted chi-square test was used to calculate the *p*-value of the characteristics of the analyzed population’s categorical variables. In the case of continuous variables, we used the Kruskal–Wallis rank-sum test to calculate the *p*-value (Table 1). Secondly, all blood metal levels were initially naturally log-transformed for further analysis because their distributions were skewed. We constructed three kinds of weighted multiple linear regression models that adjusted various variables shown in Table 2 to identify the association between the blood metal concentrations and blood eosinophil count (non-adjusted model, minimally adjusted model, and fully adjusted model). Thirdly, we constructed the machine learning XGBoost algorithm model to predict the relative importance of blood metal on the effect of blood eosinophil count (Figure 2). The XGBoost model was used to analyze blood metal contribution (gain) to blood eosinophil count. Next, we found the statistical difference between the blood lead and blood eosinophil count. Therefore, we further conducted a stratified analysis to determine the stratified association between the blood lead and blood eosinophil count through stratified multivariate logistic regression. Finally, based on the penalty spline method, we constructed a smooth curve using a generalized additive model (GAM) model with a fully adjusted model to explore the potential linear relationship between blood lead concentration and blood eosinophil count. A two-piecewise linear regression model was applied if a non-linear correlation was detected to determine the threshold effect of blood lead concentration on blood eosinophil count. When the ratio between blood lead concentration and blood eosinophil count appeared obvious in a smooth curve, the recursive method automatically calculated the inflection point, where the maximum model likelihood was used. To prevent the bias caused by missing data, we curated the NHANES database to improve the accuracy of the analysis by using the MICE package to account for missing data. We found the results of data without missing covariates were basically consistent with those of data with missing covariates and multiple imputation data. All in all, univariate and multiple analysis results were based on the calculated dataset as well as Rubin’s rules. All kinds of statistical analyses were performed by R software (Version 4.2.0) using the R package. The software EmpowerStats offered significant help in the analysis process, as well (http://www.empowerstats.com, 6 January 2023, X&Y Solutions, Inc., Boston, MA, USA). In our study, a *p*-value < 0.05 was considered statistically significant.

## 3. Results

### 3.1. Characteristics of the Participants

The baseline characteristics, which were of a weighted distribution, are shown in Table 1, including the demographic data, examination data, laboratory data, and questionnaire data of selected participants from the NHANES (2011–2018) survey. In our study, the average age of selected participants was 45.46 years old, and non-Hispanic Whites were the main population. Then, we divided different blood eosinophil counts into four quartiles (Q1–Q4). The distribution of race, education, marital status, poverty-to-income ratio, smoking status, diabetes history, blood urea nitrogen and blood cotinine in Q1–Q4 of blood eosinophil count indicated no statistical difference (*p*-value > 0.05), and gender, age, BMI, hypertension history, blood creatinine showed statistical differences (*p*-values < 0.05). Compared with the various groups, the distribution difference of blood Pb showed statistical significance, which may indicate a difference in exposure to Pb. However, blood Cd, Hg, Se, and Mn showed no exposure difference between the four quartile groups of blood eosinophil count.

### 3.2. The Associations between the Blood Metal Concentrations and Blood Eosinophil Counts

After all blood metal concentrations were naturally log-transformed, we applied the weighted multivariate linear regression model to assess the association of blood metal concentrations with blood eosinophil count in three different models (Table 2). According to the results, we found that only blood Pb was positively correlated with blood eosinophil count in the non-adjusted model and fully adjusted model, which was statistically significant. In the non-adjusted model, the blood eosinophil count increased by 24.76 (8.22, 41.30)/uL for each additional unit of blood lead of natural logarithmic conversion (ug/dL) (*p* < 0.05). In the fully adjusted model, which adjusted for gender, age, race/ethnicity, education level, marital status, poverty income ratio, BMI, smoking status, hypertension history, diabetes history, blood urea nitrogen, blood creatinine, and blood cotinine, the blood eosinophil count increased by 25.39 (6.91, 43.88)/uL for each additional unit of blood lead of natural logarithmic conversion (ug/dL) (*p* < 0.05). In the minimally adjusted model, which adjusted for gender, age, race/ethnicity, blood Pb also showed a positive association with blood eosinophil count but without statistical significance (*p* > 0.05). The above results indicated that long-time Pb exposure as an independent risk factor may increase the blood eosinophil counts of adults with asthma.

### 3.3. Stratified Associations between Blood Pb Concentrations and Blood Eosinophil Counts

To confirm the stability of multivariate linear regression analysis results, we further analyzed stratified associations between the blood Pb concentrations and blood eosinophil counts in a specific subgroup by gender, age, race, education level, marital status, poverty-to-income ratio, BMI, smoking status, hypertension history, and diabetes history (Figure 2). According to stratified analysis results, it was possible that males, age < 40 and ≥60 years old, non-Hispanic Whites, those who declared their marital status to be single, the low and middle group of poverty-to-income ratio, 25 < BMI ≤ 28, those who had smoked <100 cigarettes and ≥100 cigarettes in life, those who were without hypertension, and those who were without diabetes had higher blood eosinophil counts, with increasing blood Pb concentrations displaying a significant trend (*p* < 0.05). Furthermore, we found that the variables of gender, poverty-to-income ratio, and blood Pb may have an interaction effect associated with blood eosinophil counts (*p* for interaction < 0.05).

### 3.4. Using Machine Learning from the XGBoost Algorithm Model to Explore the Blood Metals’ Relative Importance

To identify which metal exposure affected blood eosinophil counts of adults with asthma most, we constructed the machine learning XGBoost algorithm to determine the relative importance among all selected blood metals. The blood metal variables we selected included blood Pb, Cd, Hg, Se, and Mn. In light of the results of each blood metal’s contribution according to the XGBoost model, we observed that blood Pb was the most critical variable in the blood eosinophil counts, followed by blood Se, Mn, Hg, and Cd (Figure 3). The above analysis was conducted using logarithmically transformed blood metal data. Ultimately, blood Pb, as the most relative variable, was further applied to constructing smooth curve models in our study.

### 3.5. Exploring Dose-Response Relationships of Blood Pb Concentrations with Blood Eosinophil Counts by the Generalized Additive Model (GAM)

The GAM is very sensitive to identifying linear relationships or non-linearity. To verify the reliability and stability of the analysis results, we used the GAM to explore the linear relationship between the blood Pb concentrations and blood eosinophil counts. Based on the fully adjusted model (Figure 4), we constructed a smooth fit curve to reflect the possible association. These analyses were conducted using logarithmically transformed blood Pb data. We observed the linear relationship between blood Pb concentrations with blood eosinophil counts after adjusting gender, age, race/ethnicity, education level, marital status, poverty-to-income ratio, BMI, smoking status, hypertension history, diabetes history, blood urea nitrogen, blood creatinine, and blood cotinine. In addition, we still used the segmented regression model to further verify the linear relationship of blood Pb concentrations with blood eosinophil counts (Table 3). The log-likelihood ratio test showed that *p* was more than 0.05, indicating that there was no significant difference between model 1 (one-line model) and model 2 (segmented regression model). Thus, it was more suitable to use the one-line model, and the inflection point was not statistically significant. All the above results indicated that the blood Pb concentration was linearly and positively correlated with blood eosinophil counts.

## 4. Discussion

Environmental factors were generally considered to be related to the occurrence and progression of asthma. Blood eosinophils play a crucial role in the occurrence, progression, and treatment of asthma [43,44,45,46,47]. However, according to previous studies, the results investigating the correlation between metal elements and the occurrence and progression of asthma were inconsistent [31,48,49]. Meanwhile, the studies on the impact of metal elements on blood eosinophil counts among adults with asthma are few. In this cross-sectional study, we assessed the relationship between blood metal concentrations (Pb, Cd, Hg, Mn and Se) and blood eosinophil counts among 2026 adults with asthma who participated in the NHANES survey from 2011 to 2018 (2011–2012, 2013–2014, 2015–2016, 2017–2018) in the USA. In order to rank the relative importance of exposure to different metals to blood eosinophil counts, we first constructed the machine learning XGBoost model to determine the order of selected metals. We found that blood Pb is the most important metal on blood eosinophil counts, followed by blood Se, Mn, Hg, and Cd. Next, we constructed three weighted linear models (the non-adjusted model, minimally adjusted model, and fully adjusted model) by multivariate regression analysis to determine which metal exposure might be regarded as the independent risk factor. Our study indicated that natural log-transformed blood Pb concentrations were positively associated with blood eosinophil counts in the non-adjusted model and fully adjusted model with statistical significance. In contrast, the association between blood Se, Mn, Hg, Cd, and blood eosinophil counts was not statistically significant in the three models. The above results suggested that long-time Pb exposure as a risk factor might be correlated with the immune system disorder of asthmatic adults and affect the occurrence, development, and exacerbation of asthma.

To our knowledge, Pb is a ubiquitous environmental contaminant with potential immunotoxicity [50]. Though Pb exposures in the environment, as a result of the phase-out of leaded gasoline, have been significantly reduced, Pb is still found in paint, food, and lead-containing consumer products and detected in the general population. Especially among the population of low socioeconomic status, people are increasingly concerned about the possible adverse impacts of exposure to environmental Pb [37]. The immune system is shown to be one of the most sensitive targets for lead toxicity [51]. An epidemiological study in the USA reported that Ellen M. Wells et al. observed a significant association of lead with increased eosinophils among non-Hispanic Whites with asthma after control confounding factors [52]. Additionally, another cross-sectional study involving 2447 participants also observed a positive association between blood Pb concentrations and blood eosinophil counts [34]. However, this cross-sectional study did not conduct stratified analyses in different populations to verify the stability and reliability of this association. In the same way, we found that the association between blood Pb concentrations and blood eosinophil counts demonstrated population differences among American adults with asthma. We identified the high-risk group for Pb exposure through stratified analysis, including males aged less than 40 years old and 60 years old or older, non-Hispanic Whites, those who reported their marital status as single, the low and middle group of the poverty-to-income ratio, those who had a BMI between 25 and 28, non-Hispanic blacks, those who were without hypertension, and those who were without diabetes. Furthermore, we observed a linear relationship between natural log-transformed blood Pb concentrations and blood eosinophil counts, which indicated that the disorder of the immune system might be associated with the accumulation of Pb exposures.

In our study, the selected metal exposure included Cd, Hg, Mn, and Se in addition to lead. However, few studies investigated the associations between Cd, Hg, Mn, and Se exposures and blood eosinophil count. Cd exposure can occur through contaminated water, contaminated food, smoking, and occupational settings [53,54]. In an epidemiological study, Chao-Hsin Huang et al. reported that a high concentration of Cd was associated with a high eosinophil count [34]. Hg is a ubiquitous heavy metal, which is a growing public concern owing to its wide distribution and serious adverse health impacts. Some studies have reported modulatory effects of Hg on the immune system [55,56]. In an experimental animal study, Adam M. Schaefer et al. observed an inverse relationship of blood Hg concentrations with blood eosinophils among free-ranging bottlenose dolphins [57]. Similarly, another American study demonstrated developmental exposure to environmental Hg had no correlation with blood eosinophil counts in childhood. Mn and Se are necessary nutrients, but the intake of these elements’ concentrations beyond the homeostatic capacity of the body may cause adverse effects [58]. A previous study including 2447 adults found urine Mn was not associated with blood eosinophil counts [34]. Another small case-control study found that the asthma group had lower serum Se concentrations and higher indicators of oxidative stress [42]. In our study, in three models, we observed that the associations between blood Cd, Hg, Mn, Se, and blood eosinophil counts were not statistically significant among asthmatic adults in the USA.

Compared with previous studies, our study has several advantages. First, our study provides a nationally representative, relatively large sample of asthmatic adults, which includes information on quite a few potential confounders. Second, considering that confounders may affect the results, we identify the high-risk group on the Pb exposures and verify the stability of the results through stratified analysis. Next, we use the machine learning XGBoost algorithm to determine which type of metal exposure affects blood eosinophil counts most. Then, we observed the linear relationship between blood Pb concentrations and blood eosinophil counts by constructing a smooth curve.

Nonetheless, some limitations in interpreting our results need to be taken into account. Though our study is nationally broad, most of the data are based on the American population. Data on populations of countries with less developed economies are still lacking, and heavy metal exposure may be different due to differences in national development. Secondly, this study is a cross-sectional survey, so we cannot distinguish a causal relationship. Thirdly, we cannot determine the drug use of participants. Finally, some other potential confounders, such as information on IgE-dependent sensitization, allergic diseases, parasitic diseases, and so on, may not be taken into consideration. In conclusion, we suggest constructing more predictive models between metal exposures and body biomarkers, including blood eosinophils, in the future clinical setting to guide clinical prevention and therapy.

## 5. Conclusions

In our study, we observed that blood Pb was positively and independently correlated with blood eosinophil counts among American adults with asthma. We suggested that long-time Pb exposure as a risk factor might be correlated with the immune system disorder of asthmatic adults and affect the occurrence, development, and exacerbation of asthma. Though our study did not elucidate the exact mechanism of action of Pb on the development and exacerbation of asthma, our findings would help to better identify the association of Pb and asthma from an epidemiological perspective. Finally, we wished to attract more attention to the association between metal exposure and asthma.

## Figures and Tables

**Figure 1 jcm-12-01543-f001:**
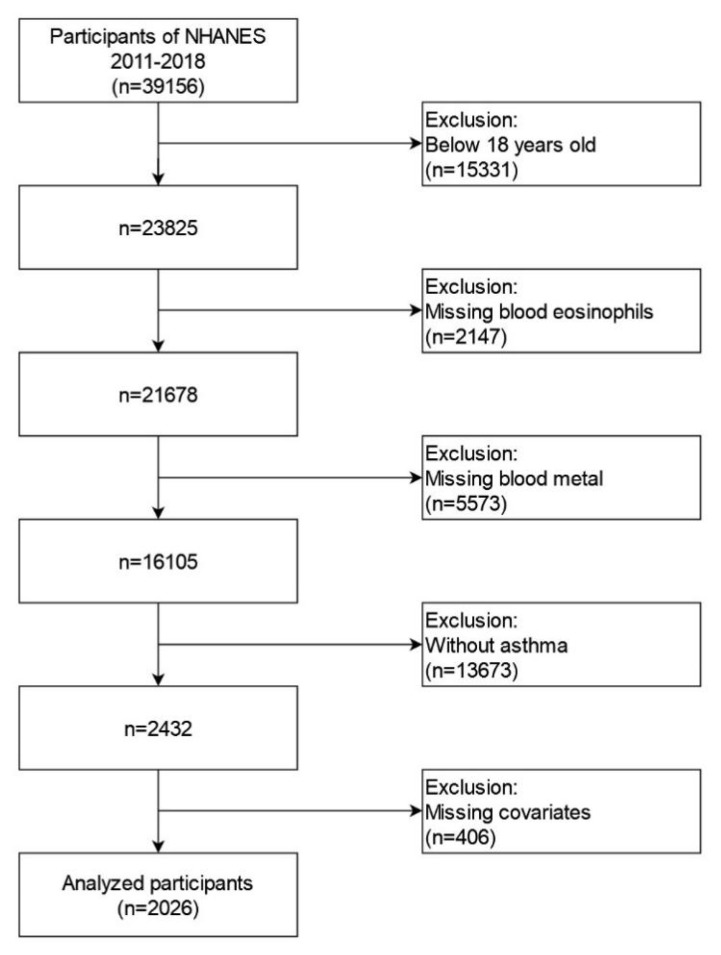
Flowchart for selecting analyzed participants.

**Figure 2 jcm-12-01543-f002:**
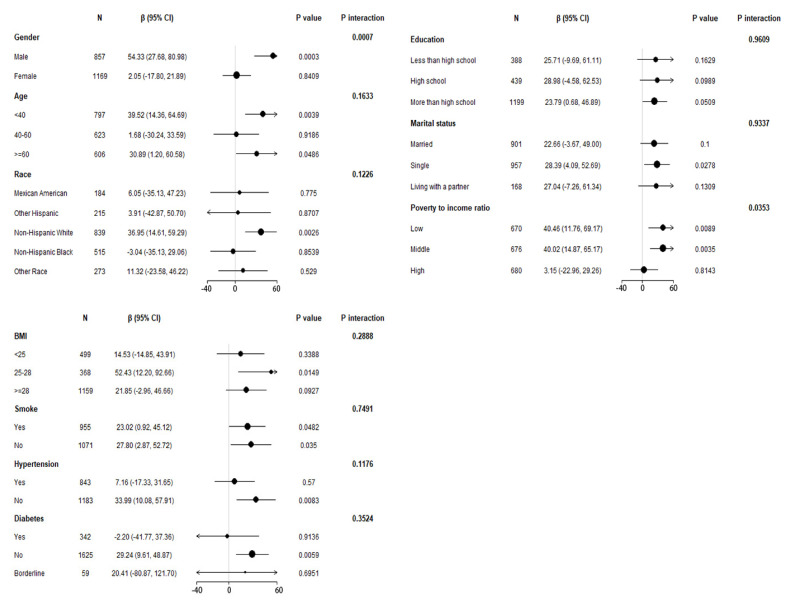
Stratified association of blood Pb on blood eosinophil counts in the prespecified and exploratory subgroups. Above adjusts for gender, age, race/ethnicity, education level, marital status, poverty to income ratio, BMI, smoked at least 100 cigarettes in life, hypertension, diabetes, blood urea nitrogen, blood creatinine and blood cotinine. In each case, the model was not adjusted for the stratification variable itself.

**Figure 3 jcm-12-01543-f003:**
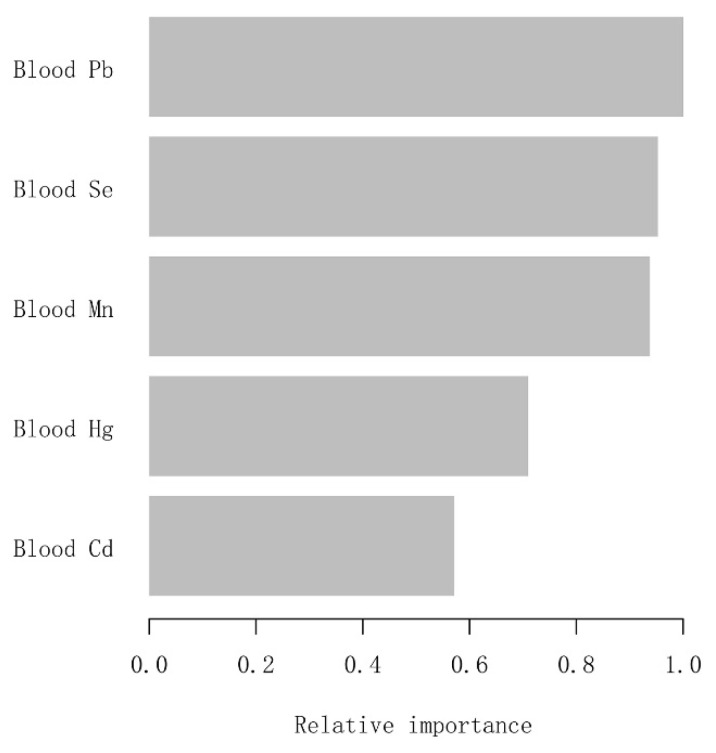
XGBoost model reveals the relative importance of each blood metal on blood eosinophil counts and the corresponding variable importance score. The *X*-axis is the importance score, the relative number of a variable used to distribute the data; the *Y*-axis is the blood metal.

**Figure 4 jcm-12-01543-f004:**
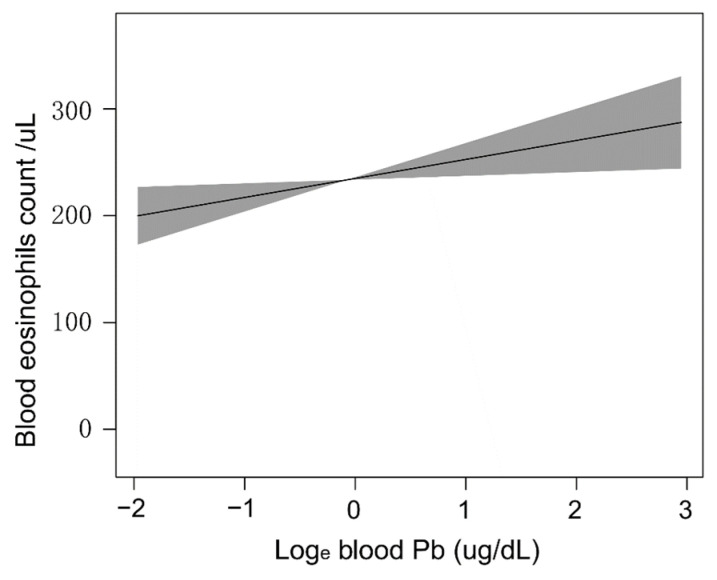
Dose-response relationships of blood Pb concentrations with blood eosinophil counts. The solid line and shadow area represented the corresponding β values and 95% confidence intervals.

**Table 1 jcm-12-01543-t001:** Clinical characteristics of the study population disaggregated by quartiles of blood eosinophil count.

	Q1	Q2	Q3	Q4	*p*-Value
Gender (%)					0.039
Male	37.41 (26.44, 49.85)	37.87 (32.63, 43.41)	39.27 (33.83, 44.98)	46.96 (42.24, 51.74)	
Female	62.59 (50.15, 73.56)	62.13 (56.59, 67.37)	60.73 (55.02, 66.17)	53.04 (48.26, 57.76)	
Age, mean (95% CI) (years)	45.46 (41.40, 49.52)	43.48 (41.89, 45.06)	45.41 (43.54, 47.28)	46.64 (44.97, 48.30)	0.038
Race/ethnicity (%)					0.361
Mexican American	4.73 (1.52, 13.79)	4.91 (3.34, 7.16)	6.13 (4.14, 8.99)	7.03 (4.96, 9.88)	
Other Hispanic	6.84 (2.44, 17.76)	6.14 (4.17, 8.96)	7.20 (5.22, 9.86)	6.56 (4.61, 9.24)	
Non-Hispanic White	63.01 (44.80, 78.14)	67.54 (60.54, 73.84)	66.28 (59.36, 72.58)	68.35 (63.38, 72.93)	
Non-Hispanic Black	20.79 (12.00, 33.56)	13.01 (9.69, 17.26)	11.36 (8.75, 14.64)	10.40 (7.98, 13.46)	
Other race	4.63 (1.79, 11.40)	8.40 (5.51, 12.61)	9.02 (6.42, 12.54)	7.66 (5.75, 10.13)	
Education (%)					0.502
Less than high school	16.70 (9.34, 28.08)	10.86 (8.44, 13.86)	14.20 (10.82, 18.42)	13.28 (9.82, 17.72)	
High school	23.31 (12.86, 38.51)	20.27 (16.29, 24.92)	19.30 (14.90, 24.63)	22.88 (18.43, 28.03)	
More than high school	59.98 (44.64, 73.59)	68.88 (63.79, 73.55)	66.49 (60.71, 71.82)	63.84 (57.45, 69.79)	
Marital status (%)					0.501
Married	45.22 (31.88, 59.29)	50.45 (45.43, 55.46)	46.76 (41.46, 52.14)	51.84 (47.29, 56.35)	
Single	50.68 (36.51, 64.74)	41.19 (36.81, 45.73)	46.01 (40.48, 51.64)	40.95 (37.12, 44.89)	
Living with a partner	4.10 (1.21, 13.00)	8.36 (5.87, 11.77)	7.22 (4.98, 10.36)	7.21 (5.06, 10.19)	
Poverty-to-income ratio, mean (95% CI)	2.60 (2.07, 3.13)	2.88 (2.68, 3.08)	2.65 (2.43, 2.88)	2.75 (2.53, 2.96)	0.241
BMI, mean (95% CI) (kg/m^2^)	28.05 (25.90, 30.20)	29.41 (28.55, 30.26)	31.03 (29.96, 32.10)	31.70 (30.80, 32.59)	0.001
Smoked at least 100 cigarettes in life (%)					0.224
Yes	50.51 (34.88, 66.05)	43.61 (37.64, 49.77)	45.57 (39.23, 52.06)	51.05 (44.87, 57.20)	
No	49.49 (33.95, 65.12)	56.39 (50.23, 62.36)	54.43 (47.94, 60.77)	48.95 (42.80, 55.13)	
Hypertension (%)					0.003
Yes	36.62 (23.68, 51.82)	29.25 (24.35, 34.68)	37.71 (32.22, 43.53)	41.20 (36.32, 46.25)	
No	63.38 (48.18, 76.32)	70.75 (65.32, 75.65)	62.29 (56.47, 67.78)	58.80 (53.75, 63.68)	
Diabetes (%)					0.394
Yes	13.66 (6.13, 27.72)	8.60 (6.62, 11.09)	12.50 (9.65, 16.04)	13.57 (10.14, 17.92)	
No	82.01 (66.25, 91.36)	88.34 (84.69, 91.22)	85.06 (80.97, 88.39)	84.24 (79.54, 88.03)	
Borderline	4.33 (0.98, 17.18)	3.06 (1.32, 6.94)	2.44 (1.26, 4.67)	2.19 (1.29, 3.70)	
BUN, mean (95% CI) (mmol/L)	4.72 (4.20, 5.24)	4.65 (4.50, 4.80)	4.79 (4.58, 5.00)	4.97 (4.80, 5.15)	0.075
Cr, mean (95% CI) (umol/L)	73.95 (69.17, 78.72)	74.05 (72.23, 75.86)	77.18 (74.93, 79.42)	80.06 (77.99, 82.13)	0.0002
Blood cotinine, mean (95% CI) (ng/mL)	94.98 (52.24, 137.72)	53.90 (38.51, 69.29)	61.44 (49.13, 73.75)	68.29 (51.47, 85.12)	0.200
Blood Pb, mean (95% CI) (ug/dL)	1.20 (0.92, 1.49)	1.00 (0.91, 1.08)	1.03 (0.94, 1.13)	1.17 (1.06, 1.28)	0.007
Blood Cd, mean (95% CI) (ug/dL)	0.56 (0.42, 0.71)	0.46 (0.37, 0.55)	0.49 (0.44, 0.54)	0.59 (0.50, 0.69)	0.106
Blood Hg, mean (95% CI) (ug/dL)	1.53 (1.00, 2.06)	1.40 (1.14, 1.65)	1.24 (1.05, 1.42)	1.13 (0.88, 1.37)	0.126
Blood Se, mean (95% CI) (ug/dL)	197.90 (190.03, 205.77)	194.14 (191.52, 196.75)	192.45 (189.50, 195.40)	194.60 (191.24, 197.97)	0.567
Blood Mn, mean (95% CI) (ug/dL)	9.62 (8.37, 10.88)	10.15 (9.81, 10.49)	10.10 (9.77, 10.42)	9.72 (9.39, 10.06)	0.179

Note: Data are expressed as weighted means (95% Cis) or proportions. Q1–Q4: Grouped by quartile according to the blood eosinophil counts. Our data included blood metal concentrations, blood eosinophil counts, demographic data, examination data, laboratory data, and questionnaire data for the second analysis. BUN: blood urea nitrogen; Cr: blood creatinine; Pb: lead; Cd: cadmium; Hg: mercury; Se: selenium; Mn: manganese.

**Table 2 jcm-12-01543-t002:** Multivariate weighted linear model analysis reveals the association between the blood metal and blood eosinophil count.

	Non-Adjusted Model	Minimally Adjusted Model	Fully Adjusted Model
	β (95% CI) *p* value	β (95% CI) *p* value	β (95% CI) *p* value
Log blood Pb (ug/dL)	24.76 (8.22, 41.30) 0.005	17.66 (−0.78, 36.09) 0.066	25.39 (6.91, 43.88) 0.010
Log blood Cd (ug/dL)	4.18 (−7.55, 15.91) 0.487	7.18 (−5.56, 19.92) 0.274	17.51 (−0.43, 35.45) 0.063
Log blood Hg (ug/dL)	−1.91 (−11.32, 7.50) 0.693	−4.35 (−14.13, 5.43) 0.387	−3.23 (−13.20, 6.75) 0.530
Log blood Se (ug/dL)	14.48 (−68.49, 97.45) 0.734	−6.99 (−86.32, 72.35) 0.864	−7.60 (−82.84, 67.64) 0.844
Log blood Mn (ug/dL)	−14.27 (−40.80, 12.26) 0.296	−6.18 (−37.67, 25.31) 0.702	−5.87 (−37.43, 25.68) 0.717

Note: Non-adjusted model adjusts for none. Minimally adjusted model adjusts for age, race/ethnicity. Fully adjusted model adjusts for gender, age, race/ethnicity, education level, marital status, poverty to income ratio, BMI, smoked at least 100 cigarettes in life, hypertension, diabetes, blood urea nitrogen, blood creatinine and blood cotinine.

**Table 3 jcm-12-01543-t003:** Threshold effect analysis of between blood Pb and blood eosinophil count using the two-piecewise linear regression model.

	β (95% CI) *p*-Value
Model I	
Linear effect	25.39 (10.53, 40.26) 0.001
Model II	
Inflection point (K)	−0.87
Log blood Pb < K	56.23 (−5.73, 118.19) 0.075
Log blood Pb > K	21.30 (4.43, 38.18) 0.013
Log likelihood ratio	0.312

Note: Models I and II all adjusted for gender, age, race/ethnicity, education level, marital status, poverty-to-income ratio, BMI, smoking status, hypertension history, diabetes history, blood urea nitrogen, blood creatinine, and blood cotinine.

## Data Availability

All data can be obtained on the NHANES official website (http://www.cdc.gov/nchs/nhanes/, 6 January 2023).

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
