# Peer review of "Association between Exposure to Selected Heavy Metals and Blood Eosinophil Counts in Asthmatic Adults: Results from NHANES 2011–2018"

_jcm, 2023, doi:10.3390/jcm12041543_

Round 1

Reviewer 1 Report

The work presented for evaluation deals with an important problem - the potential impact of heavy metals on the course of asthma.

The work needs to be corrected. Guidance was provided directly to the authors.

Recommendations for the Authors:

Introduction - I suggest moving some of the information contained in the discussion to the introduction.

Lines 77-86 are a copy of the directions to the authors. Please remove.

Factors affecting eosinophil levels: important factors such as IgE-dependent sensitization, AD, parasitic diseases and immunodeficiencies were not included. Please clarify.

Discussion-I suggest shortening, moving some of the information to the introduction section.

Conclusions-This section only presents the results obtained, they are not conclusions.

Author Response

Response to Reviewer 1 Comments

Dear Reviewer:

Thank you for your letter and for the reviewer’s comments concerning our manuscript entitled “Association between selected heavy metals exposure and blood eosinophil counts in asthmatic adults: Results from NHANES 2011–2018” (Manuscript ID: 2124040). Those comments are all valuable and very helpful for revising and improving our paper, as well as the important guiding significance to our researches. We have studied comments carefully and have made correction which we hope meet with approval. Revised portion are marked in “Track”. The main corrections in the paper and the responds to the reviewer’s comments are as flowing:

Point 1: Introduction -I suggest moving some of the information contained in the discussion to the introduction.

Response 1: We thank the reviewer very much for the comment. We have moved some of the information contained in the discussion to the introduction and revised in the text.

Point 2: Lines 77-86 are a copy of the directions to the authors. Please remove.

Response 2: We thank the reviewer very much for the comment. We have deleted the directions to the authors in the text. Thank the reviewer for the suggestion again.

Point 3: Factors affecting eosinophil levels: important factors such as IgE-dependent sensitization, AD, parasitic diseases and immunodeficiencies were not included. Please clarify.

Response 3: We thank the reviewer very much for the comment. As the reviewer suggested, we searched again in the NAHNES database and found that there were no individuals of immunodeficiencies in all the samples we analyzed, and no data on IgE-dependent sensitization, AD and parasitic diseases were found in the database from 2011 to 2018 NHANES. In the future, we will try our best to include information such as IgE-dependent sensitization, allergic diseases, parasitic diseases, allergic diseases and so on into further studies.

Point 4: Discussion-I suggest shortening, moving some of the information to the introduction section.

Response 4: We would like to thank the review experts for their comments. As the reviewer suggested, we have moved some of the information contained in the discussion to the introduction and revised in the text.

Point 5: Conclusions-This section only presents the results obtained, they are not conclusions.

Response 5: We would like to thank the review expert for the comment. In our study, we observed that blood Pb was positively and independently correlated with blood eosinophil counts among American adults with asthma. We suggested that long-time Pb exposure as a risk factor might be correlated with the immune system disorder of asthmatic adults, and affect the occurrence, development, and exacerbation of asthma. Though our study didn't elucidate the exact mechanism of action of Pb on development and exacerbation of asthma, our findings would help better to identify the association of Pb and asthma from an epidemiological perspective. Finally, we wished to attract more attention to the association between metal exposure and asthma. And we have revised in the text.

Specially, thank you for your comments.

We tried our best to improve the manuscript and made some changes in the manuscript. And here we briefly listed the changes and marked in “Track” revised paper.

We appreciate for your warm work earnestly, and hope that the correction will meet with approval.

Once again, thank you very much for your comments and suggestions.

Reviewer 2 Report

Asthma is an important clinical condition both in children and in adults.  Investigation of environmental exposures that trigger or exacerbate asthma is important.  You have cited many papers (24, 27-33, etc) showing an association between blood lead levels and blood eosinophil counts.  The additional information here is the careful comparison with additional heavy metal exposures (Cd, Hg, Se, Mn) and the extensive statistical methods applied, including XGBoost machine learning algorithm.  Using the high-quality data from NHANES (2011-2016) surveys and lab analyses is a good strategy.  

The results seem a bit overstated, that Pb "may cause immune system disorder" leading to asthma.  What is found is an association. The causal relationship remains to be investigated.  Longitudinal studies of patients with documented exposures and investigation of exacerbations might be informative. Such a population exists in Flint, Michigan, where a disastrous decision to use polluted (acidic) water from the Flint River instead of the much higher-quality Detroit River source long used produced widespread release of lead from very old plumbing systems in the city of Flint and its homes.  

Details: 

  Abstract and throughout text and tables:  p values to 4 decimal places seem  to represent excessive claims of precision in the primary values

  Introduction is missing altogether!  Instead the journal instructions have been inserted.  

Author Response

Response to Reviewer 2 Comments

Dear Reviewer:

Thank you for your letter and for the reviewer’s comments concerning our manuscript entitled “Association between selected heavy metals exposure and blood eosinophil counts in asthmatic adults: Results from NHANES 2011–2018” (Manuscript ID: 2124040). Those comments are all valuable and very helpful for revising and improving our paper, as well as the important guiding significance to our researches. We have studied comments carefully and have made correction which we hope meet with approval. Revised portion are marked in “Track”. The main corrections in the paper and the responds to the reviewer’s comments are as flowing:

Point 1: Abstract and throughout text and tables: p values to 4 decimal places seem to represent excessive claims of precision in the primary values.

Response 1: We thank the reviewer very much for the comment. We have revised the p values in the text.

Point 2: Introduction is missing altogether! Instead the journal instructions have been inserted.

Response 2: We would like to thank the review experts for their comments. We are sorry that we don’t delete the directions to the authors in the text. And we have revised in the text. Thank the reviewer for this comment again.

Point 3: The results seem a bit overstated, that Pb "may cause immune system disorder" leading to asthma. 

Response 3: We would like to thank the review expert for their comments. We suggested that long-time Pb exposure as a risk factor might be correlated with the immune system disorder of asthmatic adults and might affect the development, exacerbation, and treatment of asthma. And we have revised in the text.

Specially, thank you for your comments.

We tried our best to improve the manuscript and made some changes in the manuscript. And here we briefly listed the changes and marked in “Track” revised paper.

We appreciate for your warm work earnestly, and hope that the correction will meet with approval.

Once again, thank you very much for your comments and suggestions.

Round 2

Reviewer 1 Report

Thank you very much for the corrections.